# The Tradeoff between Maintaining Maize (*Zea mays* L.) Productivity and Improving Soil Quality under Conservation Tillage Practice in Semi-Arid Region of Northeast China

Nana Chen [1,2], Xin Zhao [3], Shuxian Dou [2], Aixing Deng [2], Chengyan Zheng [2], Tiehua Cao [3,*], Zhenwei Song [2,*] and Weijian Zhang [2]

1   College of Life Science, Yantai University, Yantai 264005, China
2   Key Laboratory of Crop Physiology and Ecology, Ministry of Agriculture and Rural Affairs, Institute of Crop Sciences, Chinese Academy of Agricultural Sciences, Beijing 100081, China
3   Institute of Agricultural Resources and Environment, Jilin Academy of Agricultural Sciences, Changchun 130033, China
*   Correspondence: caotiehua2002@163.com (T.C.); songzhenwei@caas.cn (Z.S.); Tel./Fax: +86-10-6212-8815 (Z.S.)

**Abstract:** Conservation tillage has received strong support globally to achieve food security and minimize environmental impacts. However, there are comprehensive debates on whether it can achieve the synergy between maintaining crop yields and improving soil quality. To this end, a field experiment under continuous maize (*Zea mays* L.) cropping was conducted in northeast China. The treatment included rotary tillage with straw removal (CK, conventional tillage) and rotary tillage, subsoiling tillage, and no tillage with straw retention (CR, CS, and CN, respectively). Maize yield and a set of soil physio-chemical indicators in relation with soil quality were measured during 2017 to 2021. Results showed that CN significantly reduced the maize yield by 24.9%, 23.1%, and 19.5% on average compared to that with CR, CK, and CS treatments, respectively. CN and CS significantly increased the ratio of >2 mm soil aggregates and soil geometric mean diameter (GMD) in the 0–20 cm soil layer compared those of CK and CR treatments. However, CN and CS treatments had a higher soil bulk density and soil compaction in the 0–20 cm layer compared to those with CK and CR treatments. Soil organic carbon and total nitrogen in the 0–20 cm layer under CN and CS were higher than those with CK by 5.1–15.0% and 8.5–15.7%, whereas soil $NH_4^+$ was lower by 9.1–13.9% correspondingly. CN also reduced the soil temperature during the early-growth stage of maize. Importance analysis indicated that soil temperature, bulk density, and available nitrogen were the key factors affecting maize yield. Overall, no tillage with straw mulching could improve soil stability and soil fertility but reduced maize yield. Alternatively, minimum tillage (e.g., subsoiling tillage) with straw mulching might be a suitable practice as it maintains the maize yield and improves soil quality compared to those with conventional tillage practices in the semi-arid region of northeast China in the short term.

**Keywords:** conservation tillage; straw returning; soil compaction; soil carbon and nitrogen; maize yield

## 1. Introduction

Feeding a growing global population and minimizing environmental impacts are primary challenges for crop production nowadays [1]. Sustainable intensification of crop production that aims at increasing crop yields with lower environmental impacts and does not undermine the capacity to continue producing food in the future [2] is seen as the key to food systems achieving these goals [3].

Conservation tillage, characterized as no tillage or minimum tillage combined with crop straw retention, is a sustainable agricultural practice and has received strong support globally. Previous studies have proved that conservation tillage could prevent soil erosion [4–8], increase soil water-holding capacity and water use efficiency [9,10], save energy

input [11,12], and reduce greenhouse gas emissions [13–15]. More importantly, conservation tillage has the function of improving soil quality, such as promoting the formation of soil stable aggregates [16,17], increasing soil organic carbon (SOC) and total nitrogen (TN) sequestration [18–20], and boosting the soil available nutrients [21,22]. However, studies also showed that conservation tillage did not reduce soil erosions or improve soil quality in some regions. For example, Sun et al. [23] found that conservation tillage led to soil carbon loss in cold regions. Lampurlanés and Cantero-martinez [24] reported that conservation tillage increased the soil bulk density and compaction compared to those with conventional tillage. Therefore, a more careful assessment of soil quality improvement under conservation tillage is needed.

On the other hand, the effect of conservation tillage on crop yield is not always consistent. Some studies indicated that conservation tillage increased crop yield [25–28]. Other studies also reported the opposite results that conservation tillage resulted in a crop yield reduction [29–31]. These inconsistent results may be related to specific crops, conservation tillage practices, climatic conditions, cropping regions, and socio-economic factors [32–34]. Among these studies, Li et al. [31] reported that the reduction in soil temperature from the time of sowing to emergence under no tillage with straw mulching was the primary reason for lower maize (*Zea mays* L.) yields in a high-latitude area of northeast China. Sun et al. [23] summarized that the application of conservation tillage could increase the crop yield in arid regions but not in humid regions. Corbeels et al. [32] concluded that conservation tillage was not a suitable technology to overcome low crop productivity under smallholder management in Africa, although it might bring soil conservation benefits in the short term.

The northeast China is the most important grain-production region in China, accounting for more than 30% of total maize grain production in the country [35]. Similar with other regions, the crop production in northeast China is dominated by smallholder management. Most croplands in this region are cultivated by farmers with a farm size of less than 0.6 ha, which results in the overuse of fertilizers and less agricultural income [36]. Meanwhile, continuous rotary tillage, straw removal, and small machine application in the field leads to severe soil degeneration. An investigation conducted across the northeast China indicated that the average topsoil was reduced to 13.2 cm and SOC content had decreased to 18.0 g kg$^{-1}$ over the past fifty years [35], which in turn threatens sustainable crop production. Therefore, it is urgent to optimize the agricultural management pattern to maintain the crop yield and improve soil quality. Previous studies demonstrated that conservation tillage (e.g., no tillage or subsoiling tillage with straw mulching) could be a favorable practice to realize these goals. However, whether conservation tillage can achieve the synergy between crop yield stability and soil quality improvement under smallholder management is not clear. To this end, the objectives of the present study were as follows: (1) to evaluate the effect of conservation tillage on maize yields and a set of soil physio-chemical indicators in relation to soil quality; (2) to investigate the key factors that affect maize yields. It was hypothesized that a suitable conservation tillage practice can maintain maize yields and improve soil quality. The present studies will provide scientific support for the further development of the sustainable intensification of crop production in northeast China and other similar regions in developing countries.

## 2. Materials and Methods

### 2.1. Site Description

The study was conducted during 2017 to 2021 at Shengchan Village (45°42′ N, 122°50′ E, 194 m above sea level) in Taobei District, Baicheng City, Jilin Province, China. The site is characterized as a temperate continental monsoon climate with a mean annual temperature of 6.8 °C, mean annual precipitation of 461.3 mm, of which more than 80% is distributed in June to September, annual sunshine hours of 3507.68 h, and frost-free period of 158 d. From 2017 to 2021, the mean air temperature during the maize growth period was 18.5 °C, 18.7 °C, 18.4 °C, 18.0 °C, and 18.0 °C, and the total precipitation during the maize growth period was 322.2 mm, 422.2 mm, 330.0 mm, 683.3 mm, and 490.4 mm. The monthly mean air

temperature and precipitation are presented in Figure 1. The soil is classified as Chernozem according to the FAO Soil Taxonomy. The initial soil properties at a 0–40 cm soil depth at 20 cm intervals are presented in Table 1. Before the treatment, the continuous maize cropping had been applied for more than two decades.

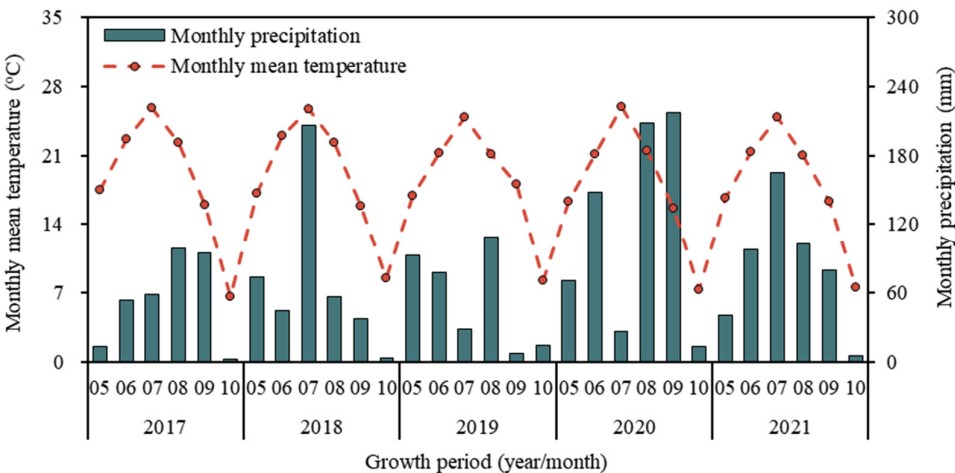

**Figure 1.** Monthly mean air temperature and precipitation during the maize growth period in 2017–2021.

**Table 1.** Initial soil properties at a 0–40 cm depth at 20 cm intervals before the experiment.

| Soil Depth (cm) | BD (g cm$^{-3}$) | SOC (g kg$^{-1}$) | TN (g kg$^{-1}$) | TP (g kg$^{-1}$) | TK (mg kg$^{-1}$) | AN (mg kg$^{-1}$) | AP (mg kg$^{-1}$) | AK (mg kg$^{-1}$) | Soil pH |
|---|---|---|---|---|---|---|---|---|---|
| 0–20 | 1.26 | 13.28 | 1.07 | 0.36 | 22.66 | 97.09 | 5.77 | 95.09 | 7.11 |
| 20–40 | 1.38 | 10.40 | 0.85 | 0.33 | 22.73 | 66.69 | 4.41 | 86.45 | 7.45 |

BD, soil bulk density; SOC, soil organic carbon; TN, soil total nitrogen; TP, soil total phosphorus; TK, soil total potassium; AN, soil available nitrogen; AP, soil available phosphorus; AK, soil available potassium.

### 2.2. Experiment Design

Four treatments were set up in fall of 2016, including rotary tillage with straw removal (the conventional tillage practice in this region, CK), rotary tillage with straw incorporation (CR), subsoiling tillage with straw mulching (minimum tillage, CS), and no tillage with straw mulching (CN). The experiment followed a randomized complete block design with three replications for each treatment. The area for each individual plot was 96 m$^2$ (10 m long × 9.6 m wide).

For CK treatment, all the above-ground biomass was removed from the field after the harvest. In the spring of next year, rotary tillage to a 20 cm soil depth and ridged seed beds with a 15 cm height with a row space of 60 cm were made before sowing. Then, the maize was planted at the top of the seed beds with a two-row conventional planter.

For CR treatment, all the straw was left in the field after the harvest and cut to about 15–20 cm. Rotary tillage to a 20 cm depth was applied to incorporate the straw into the soil evenly in the autumn. The maize was planted at a narrow space of 40 cm and wide row space of 80 cm using a two-row no-till planter. The wide and narrow space was altered every year.

For CS, all the straw was evenly distributed in the field to cover the soil surface after the harvest. In the spring of the next year, subsoiling tillage to a 30 cm depth at a wide row space was applied. Then, the same planting method as CR was applied.

For CN treatment, all the straw was evenly distributed in the field to cover the soil surface after the harvest. In the spring, maize was planted following the method in CR. No other soil tillage practice was applied.

The maize variety was Xiangyu 998 at the planting density of 67,500 plants ha$^{-1}$. Maize was sowed in early May and harvested in late September or early October every year.

The compound fertilizer (28% N, 12% $P_2O_5$, 14% $K_2O$), including 165 kg N ha$^{-1}$, 70.7 kg $P_2O_5$ ha$^{-1}$, and 82.5 kg $K_2O$ ha$^{-1}$ for each treatment were applied. Nitrogen, phosphorus, and potassium fertilizers refer to urea ($CH_4N_2O$), monocalcium phosphate ($Ca(H_2PO_4)_2$), and potassium sulphate ($K_2SO_4$), respectively. When maize is sown, the fertilizer is applied together using the same machine, and the distance between the seed and the fertilizer is maintained at about 7–10 cm. Nicosulfuron·Atrazine (a.i., 24%) and Topramzone (a.i., 30%) herbicides at the rate of 0.672 kg ha$^{-1}$ and 0.225 L ha$^{-1}$ were used to control weeds during the period of V3–V5 (third leaf blade–fifth leaf blade). The Chlorantraniliprole (a.i., 200 g/L) insecticide at the rate of 0.35 L ha$^{-1}$ was used to control pests at the 7th day after herbicides were applied.

### 2.3. Sampling and Measurements

#### 2.3.1. Soil Bulk Density

Before the maize planting for each year, the undisturbed soil samples of the 0–20 cm and 20–40 cm layers were randomly collected using metallic cores (5 cm height by 5 cm diameter) in each plot with three replications. All samples were oven-dried for at least 72 h at 105 °C to the constant weight. Then, the soil bulk density (g cm$^{-3}$) was calculated by dividing the mass of oven-drying soiling (g) by the volume of metallic cores (cm$^3$).

#### 2.3.2. Soil Compaction

Soil compaction at the 0–40 cm layer was measured before the maize planting but after land preparation in 2021. Three locations were randomly selected in each plot, and soil compaction was determined using an SC-900 digital soil compaction meter (Field Scout, spectrum Technologies, Inc., Chicago, IL, USA) at 2.5 cm intervals.

#### 2.3.3. Soil Temperature

In 2021, soil temperatures at a 5 cm soil depth were measured using JWR93-4 temperature recorders (Junwei Instruments, Co., Ltd., Hangzhou, China). Briefly, a soil temperature thermocouple was placed in the row at a 5 cm soil depth in each plot. The soil temperatures were recorded at a 30 min interval during the maize growth period. Then, the soil temperatures were computed to mean data of different maize growing stages (i.e., VE to V3, V4 to V11, V12 to R1, and R2 to R6).

#### 2.3.4. Soil Aggregation, Organic Carbon, and Total Nitrogen

In order to evaluate the effect of tillage on soil aggregate compositions and their SOC and TN contents, undisturbed soils at 0–20 cm and 20–40 cm layers were sampled in each plot. The soil samples were gently crumbled by hand and then air-dried. Samples were stored at room temperature until the analysis. Four aggregate size classes, including >2 mm, 0.25–2 mm, 0.053–0.25 mm, and <0.053 mm, were separated by a wet-sieving method using an aggregate analyzer (TPF-100, Tuopuyunnong, Zhejiang, China). The SOC and TN of different soil aggregates were measured using a TOC analyzer (multi N/C UV, Analytik Jena AG, Jena, TH, GER) and Elemental analyzer (vario PYRO cube, Elementar, Hanau, HE, GER), respectively. Soil mean weight diameter (MWD) and geometric mean diameter (GMD) were calculated as below:

$$\text{MWD} = \frac{\sum_{i=1}^{n}(M_i \times W_i)}{\sum_{i=1}^{n} M_i} \tag{1}$$

$$\text{GMD} = \exp\left[\frac{\sum_{i=1}^{n}(M_i \times \ln W_i)}{\sum_{i=1}^{n} M_i}\right] \tag{2}$$

where, $M_i$, $W_i$, and $W_{>0.25}$ refer to soil weight for different aggregate sizes (g), mean diameter for different soil aggregate sizes (mm), and mean diameter for the >0.25 soil aggregate size (g) for different aggregate sizes, respectively. Moreover, the corresponding

mean diameters for >2 mm, 0.25–2 mm, 0.053–0.25 mm, and <0.053 mm aggregate sizes were 2 mm, 1.125 mm, 0.1515 mm, and 0.053 mm, respectively.

### 2.3.5. Soil Ammonium Nitrogen and Nitrate Nitrogen Content

After maize harvesting in 2021, soil samples at 0–20 cm and 20–40 cm layers were taken. For each plot and soil layer, three randomly sampled soils were mixed into one soil sample. The soil samples were screened with a 2 mm sieve to remove the roots and stones, air-dried, and ground through a 100 mesh sieve (0.149 mm). About 5 g of frozen fresh soil, adding 50 mL 2 mol·L$^{-1}$ KCL saturated solution, was subjected to shaking in the shaking table for 1 h and suspension standing for 3–5 min after filtration to obtain the leaching liquor. Then, soil ammonium nitrogen ($NH_4^+$-N) and soil nitrate nitrogen ($NO_3^-$-N) contents were determined with a continuous-flow analyzer (AA3 Auto Analyzer 3, SEAL, Hamburg, Germany).

### 2.3.6. Yield and Yield Components

When maize matured, two rows of 5 m$^2$ of maize in each plot were collected, which were used to determine the maize yield, number of ears per ha, grain number per ear, and 100-kernel weight. The maize yield at a 14% water content was calculated.

### 2.4. Statistical Analysis

All data were examined for a normal distribution and homogeneity of variance before the analysis of variance (ANOVA). In order to evaluate the effects of tillage and year on the maize yield and soil bulk density, a general linear model of two-way ANOVA tests was conducted for the maize grain yield and its components and soil bulk density. Other data (e.g., soil compaction, soil temperature, soil aggregate, SOC, TN, soil $NH_4^+$-N, and soil $NO_3^-$-N) were analyzed using a general linear model to conduct one-way ANOVA tests. Significant differences among the treatments were determined by performing the least significant difference (LSD) test at $p < 0.05$. The relationship among yield and yield components was evaluated by performing path analysis. The predictive importance of soil properties on maize yield was conducted via automatic linear modeling, which was performed at the confidence level of 95% [37]. All analysis was conducted using SPSS for windows 25.0 (IBM Software, Chicago, IL, USA).

## 3. Results

### 3.1. Maize Grain Yield and Its Components

Tillage practice had a significant effect on the maize yield (Figure 2A). The grain yield under CN treatment was significantly lower than that under CR, CK, and CS treatments except in 2018. The five-year mean grain yields for CK, CR, CS, and CN were 11,672 kg ha$^{-1}$, 11,949 kg ha$^{-1}$, 11,154 kg ha$^{-1}$, and 8977 kg ha$^{-1}$, respectively, which indicated that grain yield under CN treatment was lower than that under CR, CK, and CS treatments by 24.9%, 23.1%, and 19.5%, respectively. Though CR treatment achieved the highest grain yield, there was no significant difference among CR, CK, and CS treatments.

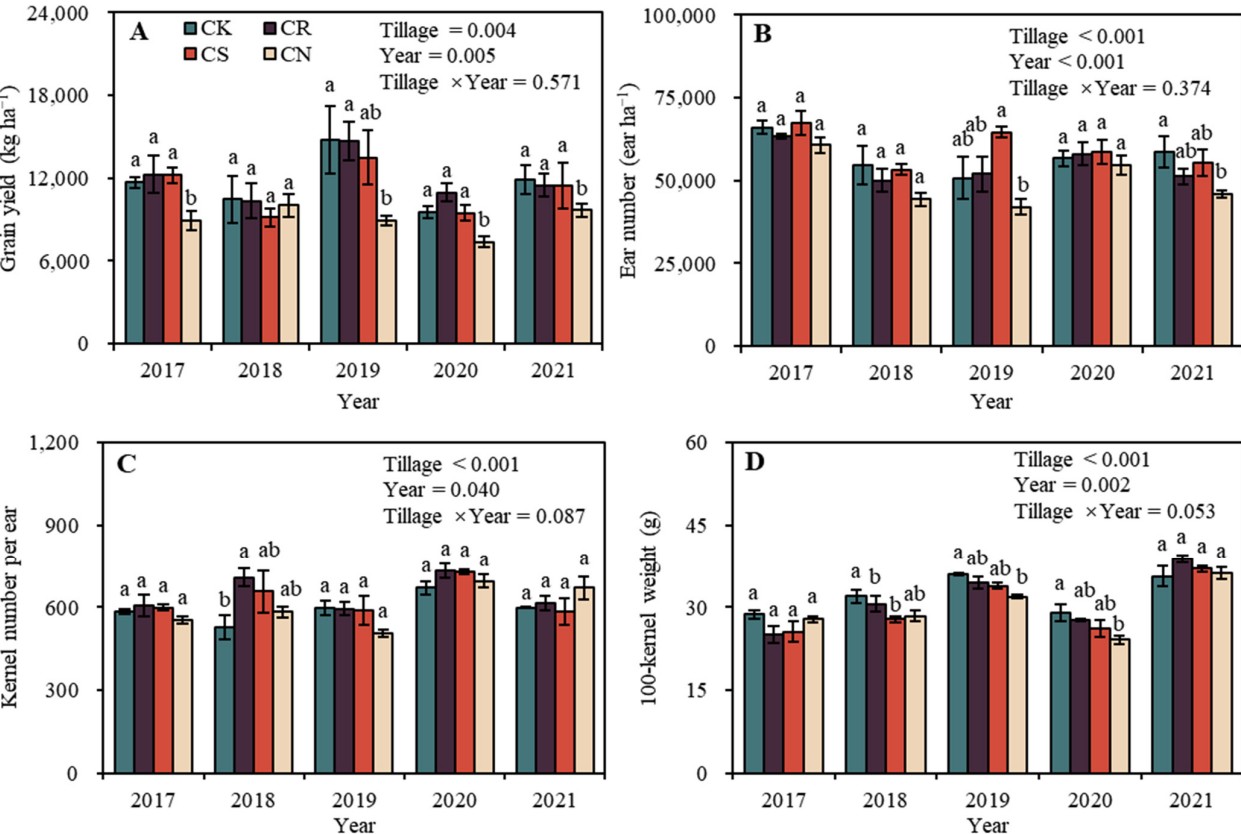

**Figure 2.** Maize yield (**A**), ear number (**B**), kernel number per ear (**C**) and 100-kernal weight (**D**) as affected by tillage regimes during 2017 to 2021. CK, rotary tillage without straw returning; CR, rotary tillage with straw incorporation; CS, subsoiling tillage with straw mulching; CN, no tillage with straw mulching. Bar means the standard error. Different letters among the treatments for each year indicate a significant difference at $p < 0.05$.

The five-year mean ear numbers under CK, CR, CS, and CN treatments were 57,333 ears ha$^{-1}$, 54,933 ears ha$^{-1}$, 59,867 ears ha$^{-1}$, and 49,533 ears ha$^{-1}$, respectively, which indicated that numbers with CN treatment were lower than those with CS, CK, and CR treatments by 17.3%,13.6%, and 9.8%, respectively (Figure 2B). However, a significant difference in ear number among the CN treatment and CK, CR, and CS treatments only occurred in 2019 and 2021. The tillage system also significantly affected the kernel number per ear (Figure 2C). The five-year mean kernel numbers per ear under CK, CR, CS, and CN treatments were 596 kernels ha$^{-1}$, 651 kernels ha$^{-1}$, 631 kernels ha$^{-1}$, and 601 kernels ha$^{-1}$, respectively, which showed that kernel number under CK treatment was lower than that under CR, CS, and CN treatments by 8.4%, 5.6%, and 0.9%, respectively. This significant difference in kernel number per ear only occurred in 2018. Figure 2D indicated that the significant difference in the 100-kernel weight among the treatments was observed in 2018 to 2020. The five-year mean 100-kernel weights under CK, CR, CS, and CN were 32.2 g, 31.3 g, 30.1 g, and 29.7 g, respectively. CN treatment resulted in 8.0%, 5.2%, and 1.3% lower 100-kernel weights than those with CK, CR, and CS treatments. It was also indicated that tillage and year had significant effects on the yield and yield components separately, while no interaction effect occurred over the periods of 2017 to 2021.

Based on the path analysis, ear number (0.581) and 100-kernel weight (0.546) had significant direct relationship with grain yield which also indirectly affected the grain yield with values of −0.135 and −0.182, respectively (Table 2).

**Table 2.** Correlation coefficient and path coefficient among grain yield and yield components (*n* = 60).

| Items | Correlation Coefficient | Direct Path Coefficient | Indirect Path Coefficient | | | |
|---|---|---|---|---|---|---|
| | | | Ear Number | Kernel Number | 100-Kernel Weight | Total |
| Ear number | 0.446 * | 0.581 * | - | 0.017 | −0.152 | −0.135 |
| Kernel number | 0.076 | 0.086 | 0.118 | - | −0.128 | −0.010 |
| 100-kernel weight | 0.364 * | 0.546 * | −0.162 | −0.020 | - | −0.182 |

\* means significant correlation at *p* < 0.05.

### 3.2. Soil Bulk Density and Compaction

The effect of the tillage system on soil bulk density presented a similar trend in the 0–20 cm soil depth each year (Figure 3A). CN treatment had a significantly higher soil bulk density by 10.5–37.0%, 11.6–36.7%, and 2.3–13.3%, respectively, compared to that with CK, CR, and CS treatments during 2018 to 2021. Meanwhile, CS treatment also increased soil bulk density by 4.9–20.9% and 6.0–20.7% compared to that with CK and CR treatments accordingly. There was no significant difference in soil bulk density under CK and CR treatments. Considering the annual variability, all treatments increased soil bulk density gradually by 26.6% (CK), 25.5% (CR), 20.3% (CS), and 8.7% (CN) during 2018 to 2021. As a result, the soil bulk density reached 1.33, 1.32, 1.53, and 1.57 g cm$^{-3}$ under the treatments of CK, CR, CS, and CN in 2021, respectively.

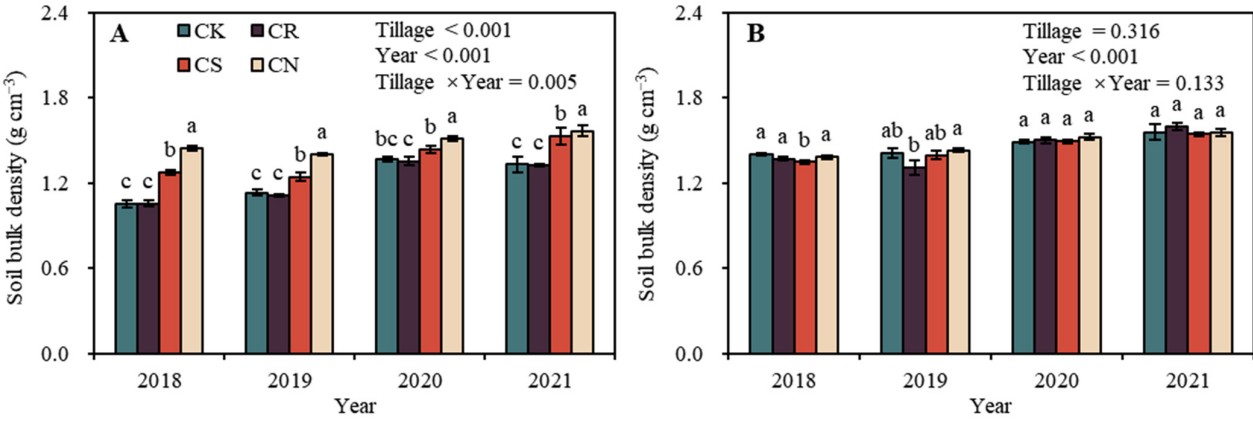

**Figure 3.** Soil bulk density at 0–20 cm (**A**) and 20–40 cm (**B**) soil depths as affected by tillage regimes during 2018 to 2021. CK, rotary tillage without straw returning; CR, rotary tillage with straw incorporation; CS, subsoiling tillage with straw mulching; CN, no tillage with straw mulching. Bar means the standard error. Different letters among the treatments for each year indicate a significant difference at *p* < 0.05.

In the 20–40 soil depth, the tillage practice showed a lesser effect on soil bulk density compared to that in the 0–20 cm soil depth (Figure 3B). Though CS treatment had a lower soil bulk density than other treatments, the significant difference only occurred in 2018 and 2019. In contrast with soil bulk density in 2018, the CK, CR, CS, and CN increased it by 11.1%, 16.6%, 14.5%, and 12.3% in 2021, respectively, which indicated that tillage had a profound impact on the deeper soil layer. In general, soil bulk density under CK, CR, CS, and CN treatments reached 1.56, 1.60, 1.54, and 1.55 g cm$^{-3}$ in 2021, respectively.

Soil compaction increased rapidly at the 0–20 cm layer and then decreased slowly at the 20–40 cm layer along with the soil depth for all treatments (Figure 4). The mean soil compaction at the 0–20 cm layer followed CS > CN > CK > CR. Moreover, CS and CN treatments were significantly higher than that of CK and CR. In the 20–40 cm layer, mean soil compaction showed a reversed trend, i.e., CR > CK > CN > CS. However, there was no significant difference in soil compaction among the treatments.

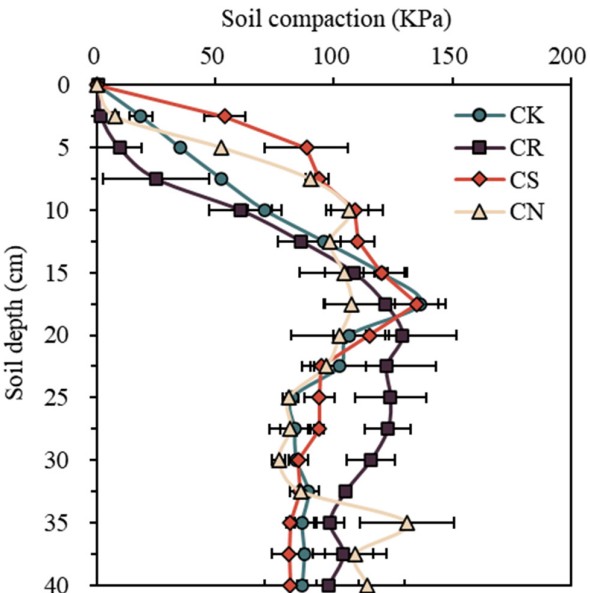

**Figure 4.** Soil compaction at the 0–40 cm depth as affected by tillage regimes in 2021. CK, rotary tillage without straw returning; CR, rotary tillage with straw incorporation; CS, subsoiling tillage with straw mulching; CN, no tillage with straw mulching. Bar means the standard error.

### 3.3. Soil Temperature during Maize Growing Period

In 2021, soil temperature in the 5 cm depth was continuously measured (Figure 5). It was shown that mean soil temperature under CN treatment was lower than that under CK, CR, and CS by 1.5, 0.5, and 0.4 °C, respectively during the emergence (VE) to third leaf (V3) stage. Then, CS treatment had the highest soil temperature during the fourth leaf (V4) to silking (R1) stage. A contrary trend occurred during the blister (R2) to physiological maturity (R6) stage, which indicated that CN increased soil temperature by 0.8, 0.4, and 0.2 °C compared to that with CK, CR, and CS treatments. However, there was no significant difference in mean soil temperature over the period of the maize growing season.

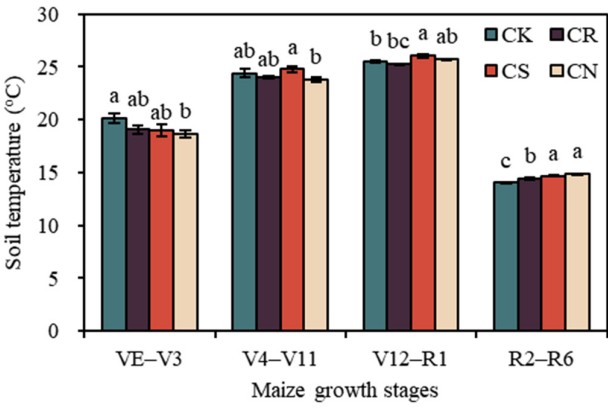

**Figure 5.** Soil mean temperature at a 5 cm soil depth as affected by tillage regimes during the maize growth period in 2021. CK, rotary tillage without straw returning; CR, rotary tillage with straw incorporation; CS, subsoiling tillage with straw mulching; CN, no tillage with straw mulching. Bar means the standard error. Different letters among the treatments indicate a significant difference at $p < 0.05$. VE, seed emergence; Vi, nth leaf; R1, silking stage; R2, blister stage; R6, physiological maturity.

### 3.4. Soil Aggregates and Soil Stability

After a five-year tillage practice, the percentage of >2 mm soil aggregates in the 0–20 cm layer under CS and CN treatments was significantly higher than that under CK and CR treatments, while CK treatment had a higher percentage of <0.053 mm soil

aggregates compared to that with other tillage systems (Figure 6A). There was no significant difference in the percentage of any soil aggregate fractions in the 20–40 cm layer (Figure 6B).

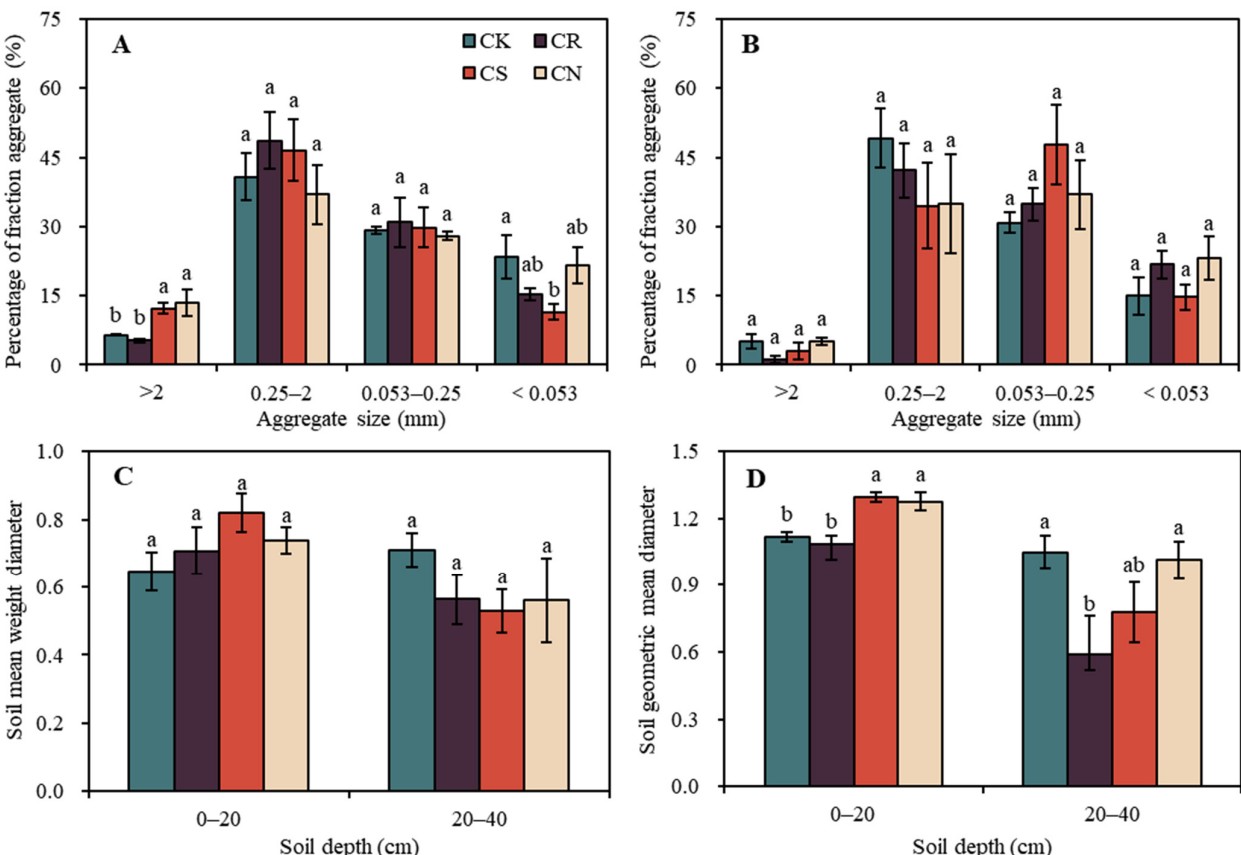

**Figure 6.** Soil aggregate fractions at the 0–20 cm (**A**) and 20–40 cm (**B**) soil depths, soil mean weight diameter (**C**), and soil geometric diameter (**D**), as affected by tillage regimes in 2021. CK, rotary tillage without straw returning; CR, rotary tillage with straw incorporation; CS, subsoiling tillage with straw mulching; CN, no tillage with straw mulching. Bar means the standard error. Different letters among the treatments for each soil aggregate fractions indicate a significant difference at $p < 0.05$.

Soil stability can be evaluated by the soil mean weight diameter (MWD) and soil geometric mean diameter (GMD). Though CS and CN treatments had a higher MWD in the 0–20 cm layer but a lower MWD in the 20–40 cm than CK and CR, there was no significant difference among the treatments (Figure 6C). However, the GMD of CN and CS treatments in the 0–20 cm layer was significantly higher than that of the CK and CR treatments by 12.5–15.8% and 15.8–19.2% (Figure 6D).

### 3.5. Soil Organic Carbon and Total Nitrogen Content

SOC and TN determine soil fertility or soil quality to a certain extent. According to Figure 7A, CN and CR treatments had higher aggregate-associated SOC contents in the >2 mm, 0.25–2 mm, and 0.053–0.25 mm fractions than CK and CS in the 0–20 cm soil layer. There were no significant differences for any aggregate fractions in the SOC content among treatments in the 20–40 cm soil layer (Figure 7B). Considering the total SOC content, CN, CR, and CS treatments were higher than the CK treatment in the 0–20 layer, but no significant difference occurred among the treatments in the 20–40 cm layer (Figure 8A).

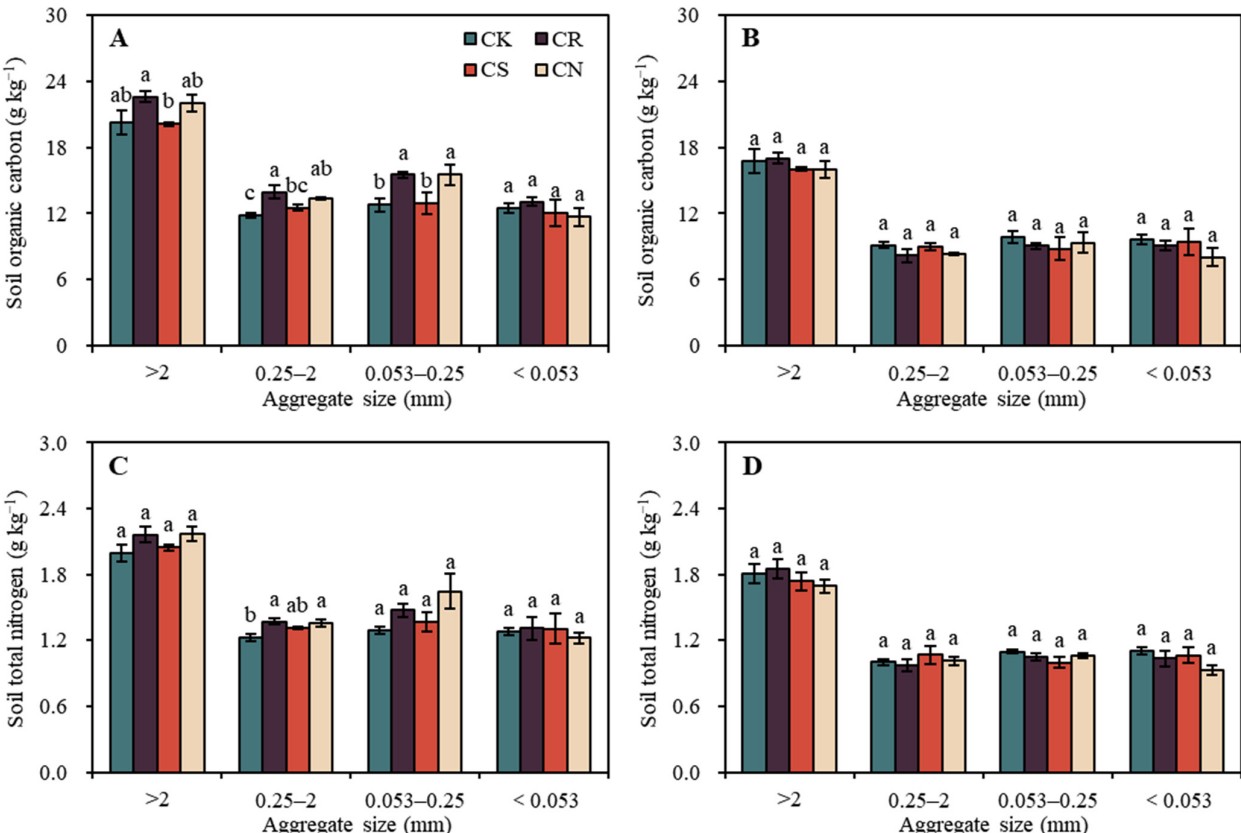

**Figure 7.** Soil organic carbon content for different soil aggregate fractions at the 0–20 cm (**A**) and 20–40 cm (**B**) soil depths and soil total nitrogen content for different soil aggregate fractions at the 0–20 cm (**C**) and 20–40 cm (**D**) soil depths as affected by tillage regimes in 2021. CK, rotary tillage without straw returning; CR, rotary tillage with straw incorporation; CS, subsoiling tillage with straw mulching; CN, no tillage with straw mulching. Bar means the standard error. Different letters among the treatments for each soil aggregate fraction indicate a significant difference at $p < 0.05$.

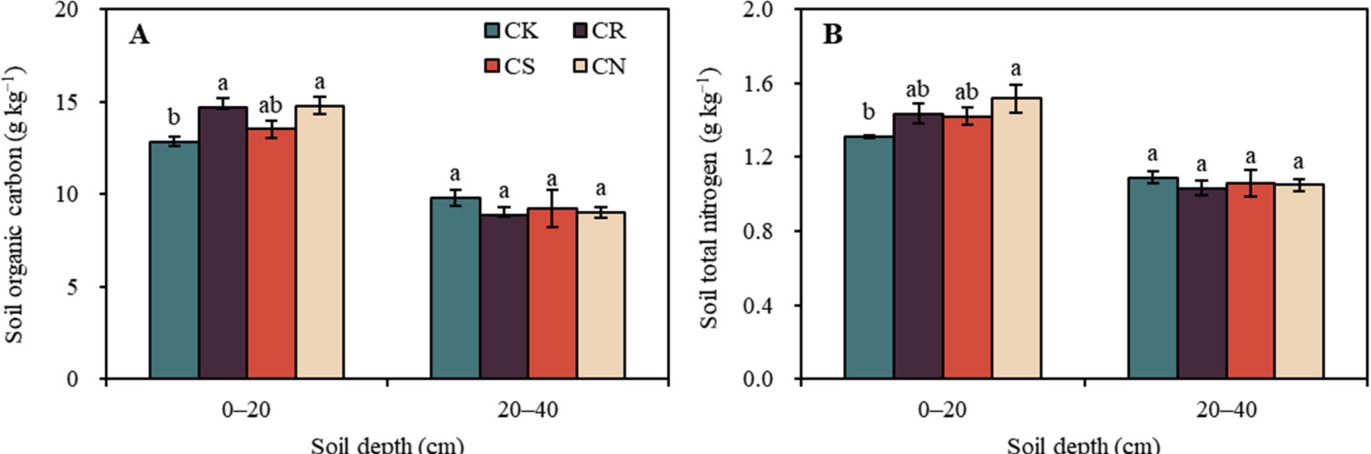

**Figure 8.** Soil organic carbon content (**A**) and soil total nitrogen content (**B**) at the 0–20 cm and 20–40 cm soil depths as affected by tillage regimes in 2021. CK, rotary tillage without straw returning; CR, rotary tillage with straw incorporation; CS, subsoiling tillage with straw mulching; CN, no tillage with straw mulching. Bar means the standard error. Different letters among the treatments for each soil aggregate fraction indicate a significant difference at $p < 0.05$.

Differences in aggregate-associated TN content in the 0–20 cm layer among the treatments were similar with those of SOC content. However, a significant difference only occurred in the 0.25–2 mm aggregate, in that CN and CR were higher than CK by 10.3% and 11.4% (Figure 7C). The content of TN in the 0–20 cm layer under CN, CS, and CR treatments was higher than that under CK by 14.6%, 8.5%, and 9.7%, respectively (Figure 8B). However, the significant difference was only tested between CN and CK treatments.

### 3.6. Soil Ammonium Nitrogen and Nitrate Nitrogen

Soil ammonium nitrogen ($NH_4^+$-N) in the 0–20 cm layer under CN and CS treatments was significant lower compared to that under the CR treatment by 23.8% and 14.2% (Figure 9). CK treatment also had significantly higher soil $NH_4^+$-N and nitrate nitrogen ($NO_3^-$-N) in the 0–20 cm layer than the CN treatment by 22.5% and 17.4%. In the 20–40 cm layer, there were no significant differences in either soil $NH_4^+$-N or $NO_3^-$-N among the treatments (Figure 9).

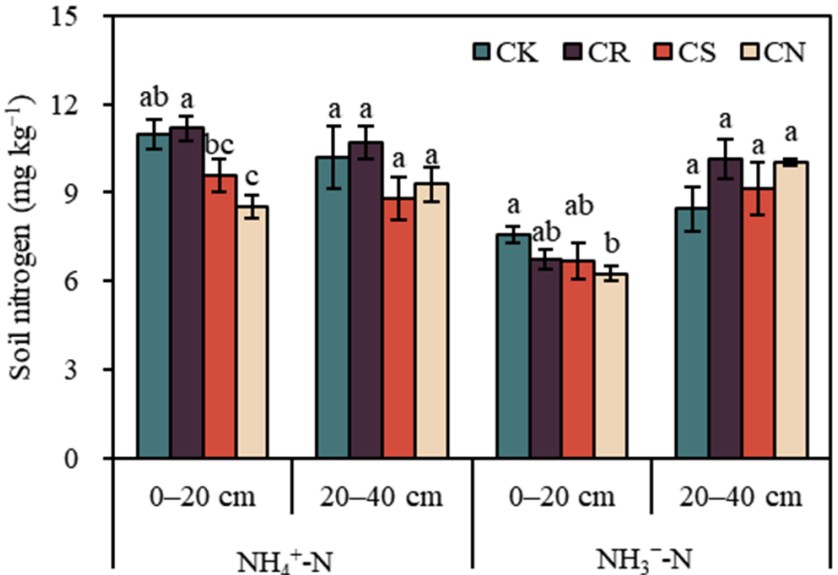

**Figure 9.** Soil $NH_4^+$-N and $NO_3^-$-N content at the 0–20 cm and 20–40 cm soil depths as affected by tillage regimes in 2021. CK, rotary tillage without straw returning; CR, rotary tillage with straw incorporation; CS, subsoiling tillage with straw mulching; CN, no tillage with straw mulching. Bar means the standard error. Different letters among the treatments indicate a significant difference at $p < 0.05$.

### 3.7. Predictive Importance of Soil Properties in Relation to Maize Yield

Based on the automatic linear modeling analysis, the most important factors affecting the grain yield were soil temperature, soil bulk density, and available nitrogen (i.e., the sum of ammonium nitrogen and nitrate nitrogen), which accounted for more than 75% of the total importance value, while other soil properties showed less importance on maize grain yields (Figure 10).

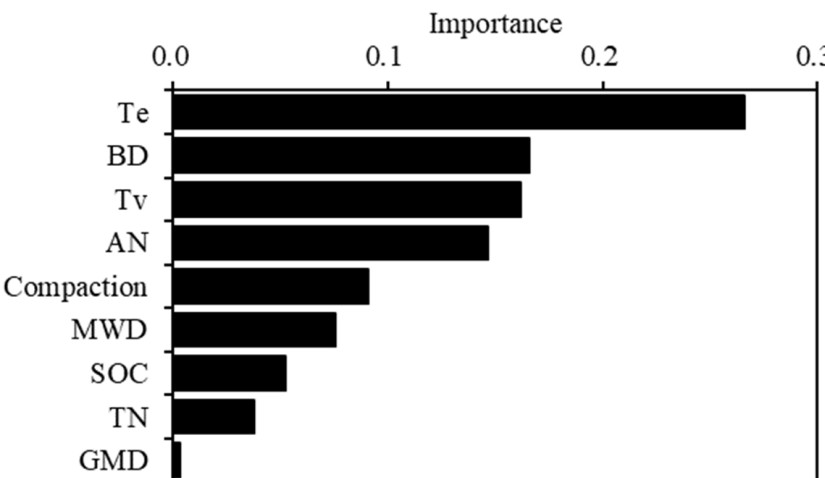

**Figure 10.** The importance of soil properties at the 0–20 cm soil depth on maize grain yield. Te, mean soil temperature during emergence to V4 of maize; BD, soil bulk density; Tv, mean soil temperature during V5 to V11 of maize; AN, available nitrogen, including ammonium nitrogen and nitrate nitrogen; Compaction, mean soil compaction; MWD, soil mean weight diameter; SOC, soil organic carbon; TN, soil total nitrogen; GMD, soil geometric mean diameter.

## 4. Discussion

### *4.1. Effect of Tillage Practice on Soil Quality*

Soil quality is defined as the capacity of soil to perform multiple functions, which is of importance to sustainable crop production in agricultural ecosystems [38,39]. A range of soil properties, including physical, chemical, and biological indicators, is used in soil quality assessments [40]. Based on the previous studies [41,42], the soil bulk density, soil compaction, soil aggregates and stability, soil organic carbon, total nitrogen, and soil available nitrogen were selected as the indicators for soil quality in the present study. The results showed that the tillage practice had a significant effect on soil quality.

#### 4.1.1. Effect of Tillage Practice on Soil Bulk and Compaction

Soil bulk density and soil compaction are physical indicators to measure soil quality, which reflects soil structure, soil permeability, soil water and fertilizer retention capacity, and resistance to root extension [33,43]. Different climate conditions, soil types, and tillage durations could lead to divergent results in soil physical properties under different tillage practices [23,30,44]. In the present study, higher soil bulk density and soil compaction were found in CN and CS treatments in the 0–20 layer, which is in accordance with Gong et al. [45]. The reason might be that no tillage and subsoiling tillage decrease the disturbance to soil and result in soil sinking naturally. Moreover, the frequent use of small tractors leads to higher stress in the soil and consequently aggravates soil bulk and compaction [46]. As a result, no tillage and subsoiling tillage may limit the absorption and utilization of water and nutrients by crop roots, affecting crop growth, development, and grain yield formation [47,48].

#### 4.1.2. Effect of Tillage Practice on Soil Aggregate and Soil Organic and Total Nitrogen

Soil aggregates are the basic unit of soil structure and the groups of SOC and TN. The formation and disruption of soil aggregates often lead to the changes in soil stability and soil carbon and nitrogen dynamics [49]. The tillage practice could affect soil aggregate composition and SOC sequestration and mineralization [50,51]. In the present study, CN and CS treatments increased the ratio of >2 mm aggregates at 0–20 cm indicating that conservation tillage is beneficial for the formation of large aggregates [16]. Furthermore, the geometric mean diameter in the 0–20 cm layer under CN and CS treatments was higher

than that under CK and CR treatments, indicating that conservation tillage improved soil stability due to reduced soil disturbances and straw returning [52].

Improved soil stability also promotes the sequestration of SOC and TN. The present study indicated that CN and CS had significant higher SOC and TN in the 0–20 cm layer. In particular, the content of SOC and TN in >0.053 mm aggregates under CN and CS treatments was much higher than that under the CK treatment, which was consistent with previous studies by Zhang et al. [53] and Zhao et al. [22].

### 4.1.3. Effect of Tillage Practice on Available Nitrogen

In the agroecosystem, the nitrogen cycle plays an important role in crop growth. Nitrogen is mainly transported from soil to the crops in the form of available mineral nitrogen, e.g., ammonium nitrogen and nitrate nitrogen [54,55]. In the present study, CN and CS treatments had lower soil ammonium nitrogen and nitrate nitrogen contents in the 0–20 cm layer than CK and CR treatments. The reason might be that the severe disturbance of rotary tillage to the soil destroyed the original structure of the soil and increased the ventilation of the soil plough layer [56]. Compared with conservation tillage, CK and CR can better integrate fertilizer into the soil, accelerate the mineralization of organic matter, enhance nitrification, and thus accumulate nitrate nitrogen [19].

### 4.2. Effect of Tillage Practice on Maize Yield

Many studies have been conducted on the effects of conservation tillage on maize yields with inconsistent results [32–34]. The present study showed that the maize yield under CN treatment was significantly lower than that under CK and CR treatments, which was consistent with Li et al. [31] who found that no tillage with straw mulching reduced the maize yield in northeast China. Previous studies reported that no tillage in the short term led to an increase in the soil bulk density, the hardening of the plough layer, the enrichment of soil organic matter in the surface layer, and the lack of nutrients in the deep layer, which resulted in the decline and instability of the crop yield [29,30,45]. Besides the increased soil bulk density and lower soil available nitrogen supply under no tillage practices, the cold climatic condition in northeast China is another important factor that hinders the maize yield. Song et al. [57] reported that the reduced soil temperature in the 5 cm layer during the early maize growth stage could result in maize grain losses at the rate of 252 kg ha$^{-1}$ per 1 °C in northeast China. Generally, the effect of conservation tillage on soil temperature is mainly reflected in the seedling stage and decline period when the crops cover the ground less, and the early effect is more important for crop growth [58]. When the temperature rises in spring, the soil temperature of no tillage with straw mulching rises slowly [59], and the soil temperature in the early maize growth stage is significantly lower than that of rotary tillage with or without straw retention. In this study, CN treatment decreased the soil temperature by 1.5 °C and 0.8 °C at the VE–V3 and V4–V12 stages compared to that with CK treatment, partially reducing the maize yield. Another study reported that in Northeast China, no tillage with straw mulching can achieve similar or higher maize yields than conventional tillage [60], which is related to planting methods [61], regional differences [47], and the duration of the experiment. Therefore, the effects of no tillage on maize yields require additional study over the long term.

### 4.3. Key Factors Affecting Maize Yield under Tillage Practices

The importance analysis showed that soil temperature at the 5 cm layer during the VE to V12 stage, soil bulk density, and available nitrogen were the most important factors affecting the maize grain yield in northeast China. However, other soil quality indicators, such as soil compaction, soil stability, SOC, and TN, showed a lesser influence on the maize grain yield. These results proved that no tillage with straw mulching could not achieve the synergy between maintaining crop yields and improving soil quality in northeast China in the short term [31,34]. Sun et al. [23] also reported that some colder regions have yield losses and soil C losses as likely as soil C gains. Though CN treatment improved soil quality

to some extent, lower grain yields and higher soil bulk density may restrain the application of the no tillage and straw mulching practice under smallholder management in northeast China [32]. Alternatively, minimum tillage, i.e., subsoiling tillage, with straw mulching could minimize the effect of low soil temperatures and bulk density on the maize yield, as well as improve soil quality, which can be the suitable conservation tillage in the semi-arid region of northeast China.

## 5. Conclusions

Based on a five-year experiment of tillage practice in northeast China, no tillage with straw mulching (CN) reduced the maize yield by 24.9%, 23.1% on average compared to that with rotary tillage with/without straw retention (CR and CK). However, minimum tillage (i.e., subsoiling tillage) with straw mulching (CS) maintained the maize yield compared to that with CR and CK treatments. CN and CS treatments also improved the soil quality to some extent. Soil temperature, soil bulk density, and available nitrogen were the key factors affecting maize yields based on importance analysis. Overall, no tillage with straw mulching could improve soil stability and soil fertility but reduce the maize yield. As a minimum tillage practice, subsoiling tillage with straw mulching might be an alternative practice in maintaining maize yields and improving soil quality under smallholder management in the semi-arid region of northeast China in the short term.

**Author Contributions:** Conceptualization, Z.S. and T.C.; Methodology, Z.S., T.C. and N.C.; Formal Analysis, N.C., X.Z., S.D. and A.D.; Investigation, N.C., X.Z. and S.D.; Data Curation, N.C., Z.S. and T.C.; Writing—Original Draft Preparation, N.C.; Writing—Review and Editing, A.D., C.Z., T.C., Z.S. and W.Z.; Visualization, N.C.; Supervision, Z.S. and T.C.; Project administration, Z.S. and T.C.; Funding Acquisition, Z.S., W.Z. and T.C. All authors have read and agreed to the published version of the manuscript.

**Funding:** This work was funded by the Agricultural Science and Technology Innovation Program of Chinese Academy of Agricultural Sciences (ASTIP, CAAS-ZDRW202202), National Natural Science Founds of China (31671642), the earmarked fund for Modern Agro-industry Technology Research System-Green manure (CARS-22-G-16), and The United Nations Development Programme (UNDP) Runtian Project (00121838).

**Institutional Review Board Statement:** Not applicable.

**Data Availability Statement:** Not applicable.

**Conflicts of Interest:** The authors declare no conflict of interest.

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
