# Peer review of "The Tradeoff between Maintaining Maize (Zea mays L.) Productivity and Improving Soil Quality under Conservation Tillage Practice in Semi-Arid Region of Northeast China"

_agriculture, doi:10.3390/agriculture13020508_

Round 1
Reviewer 1 Report
Comments are inserted in the file itself.

Author Response
Dear Reviewer:
On behalf of all co-authors, we are very grateful to you for giving us the opportunity to revise the manuscript entitled “The tradeoff between maintaining maize productivity and improving soil quality under conservation tillage practice in semi-arid region of northeast China” (Manuscript ID: agriculture-2136325). Based on the editor and reviewers’ comments, we carefully revised manuscript. The following are the responses and revisions that we have made on an item-by-item basis.
- In line 69: Avoid mentioning China, altogether three time China has been mentioned in a sentence.
Response: Thank you for your valuable comments. We have modified this sentence to “The northeast China is the most important grain production region in China, accounting for more than 30% of total maize grain production of the country.” More details are presented in line 70-71 of the revised manuscript.
- In line 106: Treatments seems very less; the minimum error degrees of freedom should be equal or more than 12.
Response: Thanks for reviewer’s question. The purpose of the present study is to evaluate the effect of conventional tillage and no/minimum tillage on crop yield and soil properties. Four treatments in this experiment were setup, including conventional tillage practice, and conventional tillage, minimum tillage (i.e. subsoiling tillage), and no tillage with straw returning. Based on this experiment, we could compare the difference of tillage practices in crop yield, soil properties, and give the optimization of tillage practice in the region. Therefore, the number of the treatments was suitable for our research. Concerning the error degrees of freedom, more error degrees of freedom could increase the significance among the treatments. In our study, the error degrees of freedom were six which was suitable for our study because we did observe the difference in crop yield, and soil properties such as soil organic carbon, soil temperature, soil bulk density, and so on among the treatments. Many published articles also used this method. Anyway, we appreciate it for reviewer’s suggestions.
- In line 120: “The wide and narrow space altered every year” What was the reason for altering it?
Response: For CR, CS, and CN, the planting method of wide row and narrow row alternation was applied. This method is conducive to ventilation and light transmission of the maize population. Furthermore, this method could divide the farmland unit into fallow area and planting area. In the second year, the fallow area is planted maize to avoid effect of root residue on maize seeding, promote straw decomposition, and improve the utilization efficiency of natural resources.
- In line 122: Wide rows of how much?
Response: The wide row space is 80 cm and narrow space is 40 cm.
- In line 123: Here also was the spacing was altered?
Response: Yes, the spacing was altered. The reason was presented in question 3.
- In line 127: If we change the row spacing, plant population will drastically vary and that will certainly influence the crop yield. How plant population was maintained uniformly?
Response: Compare to the CK treatment, the CR, CS, and CN only change the row space with wide row space of 80 cm and narrow space of 40 cm. The mean row space of wide and narrow space is 60 cm which is same as the CK treatment. Therefore, the planting density for all the treatments was same.
- In line 129: If changing plant population by altering row spacing, how similar fertilizer dose was applied. If applied in rows, then definitely 80 cm spaced plants might have got better opportunity than 40 cm.
Response: In this study, the population of the maize is same for all treatments. The maize seeding and fertilizer application were done simultaneously by machine. The distance between the seed and the fertilizer is maintained at about 7-10 cm which avoids the problems of loss and unevenness of fertilizer caused by fertilizer spread application methods. Therefore, each maize plant received the same fertilizer dose for all the treatments and the total fertilizer rate is same for all treatment. We have revised manuscript about the fertilizer application in line 141-143.
- In line 139: But after land preparation?
Response: Thanks for the reviewer. We have modified this sentence as “Soil compaction at 0-40 cm layer was measured before the maize planting but after land preparation in 2021” in line 156-157.
- In line 143: Soil compaction and soil temperature were only recorded during 2021?
Response: In the present study, the soil compaction and soil temperature were recorded in 2021. One reason is that we measured these parameters at 5th year of the experiment which could ensure the measurement are reliable. The other reason is we didn’t measure these parameters in 2020 and 2022 due to COVID-19 pandemic. We will continuously measure these soil parameters in the future research.
- In line 176: Net plot is less, although gross plot size was about 96 m2?
Response: In this study, we selected the relatively even sub-plot of 5 m2 for each plot to determine the grain yield. We appreciate it for reviewer’s suggestion and will enlarge the sampling area in the future research.
- In line 228: What is NT, it is not used under evaluation, mention appropriate abbreviation?
Response: NT means no tillage treatment. We have replaced the NT with CN to stay same abbreviation of no tillage in the manuscript.
- In line 250: “decreased slowly or increased slightly” use one term?
Response: We have revise the sentence as “…and then decreased slowly at 20-40 cm layer along with soil depth for all treatments….” in line 271.
- In line 256: (Figure 3. Soil compaction) Use of color figure might have given more clarity. Tracking each line is difficult.
Response: Thank you for your suggestions. We have revised all figures as color figures in the manuscript.
- In line 261: (Soil temperature during maize growing period) Why only in 2021 and not in other years?
Response: In the present study, we measured the soil temperature in 2021. One reason is that we measured it at 5th year of the experiment which could ensure the measurement are reliable. The other reason is we can not measure soil properties in 2020 and 2022 due to COVID-19 pandemic. We will continue to measure soil temperature in the future research.
- In line 262: NT?
Response: We have replaced NT with CN.
- In line 269: (Figure 4. Soil mean temperature) Kindly reanalyze statistically, some treatments difference of treatments are not more than 0.5℃, but still showing significant difference.
Response: Thanks for your suggestion. The statistical analysis has been re-done. The problem that the difference among the treatments is small but shows significant differences is because the error of the repetitions of each treatment is small, which makes the treatments show significant difference.
- In line 288: (Figure 5. Soil aggregate fractions) Fraction
Response: We apologize for this word spelling mistake. We have corrected it in the figures.
- In line 452: (2. Garnett, T.; Appleby, M.C.; Balmford, A.; Bateman, I.J.; Benton, T.G.; Bloomer, P.; Burlingame, B.; Dawkins, M.; Dolan, L.; Fraser, D.; Herrero, M.; Hoffmann, I.; Smith, P.; Thornton, P.K.; Toulmin, C.; Vermeulen, S.J. Sustainable Intensification in Agriculture: Premises and Policies. Science. 2013, 341, 33-34. https://doi.org/10.1126/science.1234485.) small case
Response: We have revised the tittle as “…Sustainable intensification in agriculture: premises and policies…” in line 488-489, and have re-examined the case format of the words in the references.
Reviewer 2 Report
The article entitled, "The tradeoff between ... northeast China" dealt with important information on conservation agril. practices on maize yield, soil properties, etc. I would like to suggest the following recommendations for this article:
1. In the abstract, the authors mentioned that the no-tillage was best for soil fertility, however, reduced the yield; while the reduced tillage was best for yield. Such a recommendation is misleading. I would suggest the author to recommend like minimum (reduced) tillage be best for productivity with maintaining the soil health. As it improved the SOC than continuous tillage and not very less than no-tillage.
2. In Line 45: The authors are suggested to include the following recent reference along with reference no. 4: https://doi.org/10.3390/agronomy12112766
3. The last paragraph of the introduction should include the present research gap, the hypothesis of the study, and a novelty statement for this study.
4. The experiment was designed in RBD with three replications and four treatments. That means the df (degrees of freedom) of the study was 6. I would like to know, why the authors choose so low df. The statistical error of the study was so high. I would like to know the proper justification for this.
5. Line no 182: The author mentioned that the ANOVA was two-way. How did they use two-way ANOVA for RBD design? For RBD, the ANOVA will always be one-way. Two-way ANOVA is used for factorial experiments. Please clarify this.
6. The results are well described with the proper illustration of the figures and tables.
7. Along with reference 37, I would suggest the authors to include the following reference: https://doi.org/10.3390/agronomy11112190
8. The author should discuss well why the no-tillage reduced the maize yield. There are many previous studies that suggest that long-term no-tillage with residue incorporation ultimately improves the yield of the crop.
9. Mention a concrete recommendation for the maize growers from this study focusing on the principles of sustainable agriculture.
Author Response
Dear Reviewer:
On behalf of all co-authors, we are very grateful to you for giving us the opportunity to revise the manuscript entitled “The tradeoff between maintaining maize productivity and improving soil quality under conservation tillage practice in semi-arid region of northeast China” (Manuscript ID: agriculture-2136325). Based on the editor and reviewers’ comments, we carefully revised manuscript. The following are the responses and revisions that we have made on an item-by-item basis.
- In the abstract, the authors mentioned that the no-tillage was best for soil fertility, however, reduced the yield; while the reduced tillage was best for yield. Such a recommendation is misleading. I would suggest the author to recommend like minimum (reduced) tillage be best for productivity with maintaining the soil health. As it improved the SOC than continuous tillage and not very less than no-tillage.
Response: Many thanks for reviewer’s suggestion. We have revised the abstract as “Alternatively, minimum tillage (e.g. subsoiling tillage) with straw mulching might be a suitable practice as it maintains maize yield and improves soil quality compared with conventional tillage practice in semi-arid region of northeast China in short-term.” in line 31-33. We also replaced the reduced tillage with minimum tillage throughout the manuscript.
- In Line 45: The authors are suggested to include the following recent reference along with reference no. 4:
Response: Many thanks for reviewer’s suggestion. We have added this literature: Pramanick, B.; Kumar, M.; Naik, B.M.; Kumar, M.; Singh, S.K.; Maitra, S.; Naik, B.S.S.S.; Rajput, V.D.; Minkina, T. Long-term conservation tillage and precision nutrient management in maize–wheat cropping system: Effect on soil properties, crop production, and economics. Agronomy. 2022, 12, 2766. https://doi.org/10.3390/agronomy12112766.
- The last paragraph of the introduction should include the present research gap, the hypothesis of the study, and a novelty statement for this study.
Response: It is a good suggestion. We have revise the manuscript, and added hypothesis of this study in the Introduction “It is hypothesized that suitable conservation tillage practice can maintain maize yield and improve soil quality.” in line 87-88.
- The experiment was designed in RBD with three replications and four treatments. That means the df (degrees of freedom) of the study was 6. I would like to know, why the authors choose so low df. The statistical error of the study was so high. I would like to know the proper justification for this.
Response: Thanks for reviewer’s question. The purpose of the present study is to evaluate the effect of conventional tillage and no/minimum tillage on crop yield and soil properties. Four treatments in this experiment were setup, including conventional tillage practice, conventional tillage, minimum tillage (subsoiling tillage), and no tillage with straw returning. Based on this experiment, we could compare the difference of tillage practices in crop yield and soil properties, and give the optimization of tillage practice in the region. Therefore, the number of the treatments was suitable for our research. Basically, more error degrees of freedom could increase the significance among the treatments. In our study, the degrees of freedom were six which was suitable for our study because we did observe the difference in crop yield, and soil properties such as soil organic carbon, soil temperature, soil bulk density, and so on among the treatments. We also check the analysis data that the high error is from both treatments and block groups. We appreciate it for reviewer’s suggestion.
- Line no 182: The author mentioned that the ANOVA was two-way. How did they use two-way ANOVA for RBD design? For RBD, the ANOVA will always be one-way. Two-way ANOVA is used for factorial experiments. Please clarify this.
Response: Many thanks for reviewer’s question. The experiment was design as RBD to evaluate the tillage treatment effects on soil properties and crop yield. However, the climate condition, the duration of the tillage practice also have potential effect on the soil properties and crop yield. Therefore, we use the general linear model for two-way ANOVA tests to evaluate how tillage treatment and year affect maize yield and it components, and soil bulk density. For other parameters, we use one-way ANOVA tests to evaluate the differences among the tillage treatments. We have revised the manuscript to clarify this issue “In order to evaluate the effects of tillage and year on maize yield and soil bulk density, a general linear model of two-way ANOVA tests was conducted for maize grain yield and its components, and soil bulk density.” in line 197-204.
- Along with reference 37, I would suggest the authors to include the following reference.
Response: Many thanks for reviewer’s suggestion. We have added this literature: Laik, R.; Kumara, B.H.; Pramanick, B.; Singh, S.K.; Nidhi.; Alhomrani, M.; Gaber, A.; Hossain, A. Labile soil organic matter pools are influenced by 45 years of applied farmyard manure and mineral nitrogen in the wheat–pearl millet cropping system in the sub-tropical condition. Agronomy. 2021, 11, 2190. https://doi.org/10.3390/agronomy11112190.
- The author should discuss well why the no-tillage reduced the maize yield. There are many previous studies that suggest that long-term no-tillage with residue incorporation ultimately improves the yield of the crop.
Response: We have added the following in the manuscript for a more detailed explanation of why short-term no-tillage reduces yield: “Previous studies reported that no-tillage in short-term led to the increase of soil bulk density, the hardening of the plough layer, the enrichment of soil organic matter in the surface layer, and the lack of nutrients in the deep layer, which resulted in the decline and instability of crop yield [29,30,45].” in line 422-425.
In line 437-441, we add the following: “Another study reported that in Northeast China, no tillage with straw mulching can achieve similar or higher maize yield than conventional tillage [60], which is related to planting methods [61], regional differences [62], and duration of the experiment. Therefore, the effects of no-tillage on maize yield require additional study over the long term.”
We have also added relevant literature: Fan, R.Q.; Zhang, X.P.; Liang, A.Z.; Shi, X.H.; Chen, X.W.; Bao, K.S.; Yang, X.M.; Jia, S.X. Tillage and rotation effects on crop yield and profitability on a Black soil in Northeast China. Can. J. Soil Sci. 2012, 92, 463-470. https://doi.org/10.4141/cjss2010-020.
Kumar, S.; Kadono, A.; Lal, R.; Dick, W. Long-term no-till impacts on organic carbon and properties of two contrasting soils and corn yields in Ohio. Soil Sci. Soc. Am. J. 2012, 76, 1798-1809. https://doi.org/10.2136/sssaj2012.0055.
Li, J.Y.; Li, Q.; Wu, X.P.; Wu, H.J.; Song, X.J.; Zhang, Y.Q.; Liu, X.T.; Ding, W.T.; Zhang, M.N.; Zheng, F.J. Regional differences in effects of no-tillage on soil water holding characteristics and organic carbon storage in farmland. Sci. Agric. Sin. 2020, 53, 3729-3740. (In Chinese) doi: 10.3864/j.issn.0578-1752.2020.18.009.
- Mention a concrete recommendation for the maize growers from this study focusing on the principles of sustainable agriculture.
Response: We have adjusted to the text to be clearer, with the following modifications: “As a minimum tillage practice, subsoiling tillage with straw mulching might be an alternative practice in maintaining maize yield and improving soil quality under smallholder management in semi-arid region of northeast China in short-term.” in line 466-468.
Reviewer 3 Report
Dear authors,
your paper is very good ant the topic is interesting. The manuscript is well structured and organized, according to the journal directions. Introduction is plenty with relevant publications and very well-built. The methodology of setting up experiments and implementation is consistent with other research. The strength of the work is also the large number of results. The article requires only minor changes:
1. If the authors used plant protection products (herbicides, fungicides, etc.) in the experiment, please provide their active ingredients, doses and developmental phases of the plant in which they were used.
2. Authors should expand the discussion of research results (especially chapters 4.1.1 and 4.1.3) and quote more literature. There are many articles in the literature on the effects of tillage practice on soil bulk and compaction and on nitrogen availability.
3. In my opinion, air temperature and precipitation data are better presented in the form of a figure rather than a table.
4. The article is missing the Latin name of the Maize species.
5. In what form were fertilizers applied to maize? Were they single-component or multi-component fertilizers? This should be supplemented.
Author Response
Dear Reviewer:
On behalf of all co-authors, we are very grateful to you for giving us the opportunity to revise the manuscript entitled “The tradeoff between maintaining maize productivity and improving soil quality under conservation tillage practice in semi-arid region of northeast China” (Manuscript ID: agriculture-2136325). Based on the editor and reviewers’ comments, we carefully revised manuscript. The following are the responses and revisions that we have made on an item-by-item basis.
- If the authors used plant protection products (herbicides, fungicides, etc.) in the experiment, please provide their active ingredients, doses and developmental phases of the plant in which they were used.
Response: Thanks for reviewer’s question. We have revised the manuscript as following “Nicosulfuron·Atrazine (a.i. 24%) and Topramzone (a.i. 30%) herbicides at the rate of 0.672 kg ha-1 and 0.225 ha-1 were used to control weeds during the period of V3-V5 (third leaf blade-fifth leaf blade). The Chlorantraniliprole (a.i. 200g/L) insecticide at the rate of 0.35 kg ha-1 was used to control pests at 7th day after herbicides was applied.” More details are listed in line 143-147.
- Authors should expand the discussion of research results (especially chapters 4.1.1 and 4.1.3) and quote more literature. There are many articles in the literature on the effects of tillage practice on soil bulk and compaction and on nitrogen availability.
Response: Thanks for your suggestion, we have added the following to the manuscript for a more detailed explanation:
In line 381: Different climate conditions, soil types, tillage duration could lead to divergent results in soil physical properties under different tillage practices [23,30,44].
In line 411: The reason might be that the severe disturbance of rotary tillage to the soil destroyed the original structure of the soil and increased the ventilation of the soil plough layer [56].
We have also added relevant literature: Sun, Y.H.; Zhang, H.J.; Cheng, J.H.; Wang, Y.J.; Shi, J.; Cheng, Y. Soil characteristics and water conservation of different forest types in Jinyun Mountain. J. Soil Water Conserv. 2006, 20, 106-109. (In Chinese) DOI:10.13870/j.cnki.stbcxb.2006.02.026.
Lv, Q.S.; Zhou, B.; Wang, P. Effects of no-tillage on root properties and yield of maize and soil physical properties: an integrated analysis. J. Ecol. 2020, 39, 3492-3499. DOI:10.13292/j.1000-4890.202010.002.
Booth, M.S.; Stark, J.M.; Rastetter, E. Controls on nitrogen cycling in terrestrial ecosystems: a synthetic analysis of literature data. Ecol. Monogr. 2005, 75, 139-157. https://doi.org/10.1890/04-0988.
Zuber, S.M.; Behnke, G.D.; Nafziger, E.D.; Villamil, M.B. Crop rotation and tillage effects on soil physical and chemical properties in Illinois. Agronomy. 2015, 107, 971-978. https://doi.org/10.2134/agronj14.0465.
- In my opinion, air temperature and precipitation data are better presented in the form of a figure rather than a table.
Response: Thank you for your suggestions. We have replaced the Table 1 with Figure 1. We also added the explanation about the climate conduction in the manuscript as “From 2017 to 2021, the mean air temperature during the maize growth period was 18.5℃, 18.7℃, 18.4℃, 18.0℃, and 18.0℃, and the total precipitation during the maize growth period was 322.2 mm, 422.2 mm, 330.0 mm, 683.3 mm and 490.4 mm. The monthly mean air temperature and precipitation are presented in Figure 1.” in line 99-102.
- The article is missing the Latin name of the Maize species.
Response: Many thanks for reviewer’s suggestion. We have added the Latin name of maize in the tittle of the manuscript: The tradeoff between maintaining maize (Zea mays L.) productivity and improving soil quality under conservation tillage practice in semi-arid region of northeast China.
- In what form were fertilizers applied to maize? Were they single-component or multi-component fertilizers? This should be supplemented.
Response: Thanks for reviewer’s suggestion. We have revised the manuscript “The compound fertilizer (N-P2O5-K2O, 28%-12%-14%) including 165 kg N ha-1, 70.7 kg P2O5 ha-1, and 82.5 kg K2O ha-1 for each treatment were applied. Nitrogen, phosphorus, and potassium fertilizers refer to urea (CH4N2O), monocalcium phosphate (Ca(H2PO4)2), and potassium sulphate (K2SO4) respectively. When maize is sown, the fertilizer is applied together using the same machine, and the distance between the seed and the fertilizer is maintained at about 7-10 cm.” More details are presented in line 138-143.
Round 2
Reviewer 2 Report
The authors incorporated all the comments made by me and the paper now looks good. It may be accepted now.